# Machine learning prediction of combat basic training injury from 3D body shape images

Steven Morse[1], Kevin Talty[1], Patrick Kuiper[1], Michael Scioletti[1], Steven B. Heymsfield[2], Richard L. Atkinson[3], Diana M. Thomas[1] *

1 United States Military Academy, West Point, New York, United States of America, 2 Pennington Biomedical Research Center, Baton Rouge, Louisiana, United States of America, 3 Division of Endocrinology and Metabolism, Virginia Commonwealth University School of Medicine, Richmond, Virginia, United States of America

* diana.thomas@westpoint.edu

## Abstract

### Introduction

Athletes and military personnel are both at risk of disabling injuries due to extreme physical activity. A method to predict which individuals might be more susceptible to injury would be valuable, especially in the military where basic recruits may be discharged from service due to injury. We postulate that certain body characteristics may be used to predict risk of injury with physical activity.

### Methods

US Army basic training recruits between the ages of 17 and 21 (N = 17,680, 28% female) were scanned for uniform fitting using the 3D body imaging scanner, Human Solutions of North America at Fort Jackson, SC. From the 3D body imaging scans, a database consisting of 161 anthropometric measurements per basic training recruit was used to predict the probability of discharge from the US Army due to injury. Predictions were made using logistic regression, random forest, and artificial neural network (ANN) models. Model comparison was done using the area under the curve (AUC) of a ROC curve.

### Results

The ANN model outperformed two other models, (ANN, AUC = 0.70, [0.68,0.72], logistic regression AUC = 0.67, [0.62,0.72], random forest AUC = 0.65, [0.61,0.70]).

### Conclusions

Body shape profiles generated from a three-dimensional body scanning imaging in military personnel predicted dischargeable physical injury. The ANN model can be programmed into the scanner to deliver instantaneous predictions of risk, which may provide an opportunity to intervene to prevent injury.

**Data Availability Statement:** The data needs to be requested and authorized by release from the United States Army. Requests can be made to the Commanding General at Fort Jackson. Fort

Jackson has a contact email which will lead to the CG: usarmy.jackson.93-sig-bde.mbx.atzj-pao@mail.mil Author Diana Thomas will serve as an additional point of contact to facilitate data access. You can reach Dr. Thomas at diana.thomas@westpoint.edu.

**Funding:** The author(s) received no specific funding for this work.

**Competing interests:** The authors have declared that no competing interests exist.

## Introduction

The United States Army (US Army) anticipates basic training recruits (BTRs) will be injured during training. Most of these injuries will heal with rest, but some are more severe, leading to expensive treatments and medical separation from the US Army [1]. Injuries, such as stress fractures [2, 3], can result in discharges from the US Army basic combat training. Femoral neck injuries are also observed during basic training and can often result in discharge [4].

These injuries, some of which are situational and others of which are stress-related (or "overuse injuries" [5]), have high medical costs and can impact future quality of life for the BTR. Additionally, under the US Army's medical policy, the US Army may be financially responsible for the injured BTR's long-term care. Thus identifying BTRs at increased risk for dischargeable injury prior to training is critical [6–8].

Low physical fitness, low and high body mass index (BMI), and anthropometry are some of the well-established predictors of Army related injuries [6, 9–14]. While, for example, categories defined by BMI of injury risk would render too many false positives identified by high or low BMI to be a feasible screening mechanism. Screening BTRs with more manually obtained measurements is not currently feasible, in part due to the added burdens of measuring predictive model input variables and then delivering model predictions quickly and efficiently for the high volume of BTRs that continuously arrive at basic training sites like Fort Jackson, SC.

Recently, for the purpose of personalized uniform fitting, Fort Jackson adopted a 3D whole body scanner that provides 161 anthropometric measurements [15–17] of the body. The device images each individual recruit within seconds using laser technology and multiple cameras. The raw image data is automatically transformed to reproducible and highly precise body posture, length, and circumference measurements. In this study we utilized Fort Jackson 3D body image data which included records of individuals who eventually were separated from the service due to injury to develop machine learning models that identify body shape characteristics that correlated with injury. The best of these models was programmed into the body scanner which allows for automatic deployment of risk predictions during uniform sizing without additional burden to the current process With the advancement of 3D body image scanning technology, our approach can be extended to other sports that observe similar injuries.

## Methods

### Study oversight and ethics

The models were developed as a secondary analysis of a large de-identified dataset. The protocol for this study was determined as not constituting human subjects research by the United States Military Academy Institutional Review Board.

### Study design

We tested the hypothesis that BTRs who are discharged due to injury can be identified from body shape measurements obtained from a 3D body image scanner. We developed logistic regression, random forest and artificial neural network models [18] that use 161 different individual anthropometric measurements as model inputs and output a probability of sustaining a dischargable injury.

For all model development and analysis, we used the Python programming language (Python Software Foundation. Python Language Reference, version 2.7, https://www.python.org/).

## Human Solutions 3D body image scans

Three-dimensional scans were performed on BTRs using the 3D body imaging scanner, Human Solutions of North America (Mooresville, NC) Vitus Smart XXL 3D. Each scan took less than 20 seconds to capture a body surface image from which the machine's software automatically delivers 161 different measurements. Previous research has demonstrated an increased accuracy of these automated results in comparison with manually obtained measurements [15, 17].

## Participants

US Army BTRs (N = 17,680, 27.5% female) were scanned for uniform fitting using the 3D body imaging scanner, Human Solutions of North America (Mooresville, NC) Vitus Smart XXL 3D body scanner at Fort Jackson, SC from February 1, 2017 to October 27, 2017.

The data from these scans was paired to the discharge status of each individual BTRs that sustained injuries during the 11-week basic training program (N = 147). This consolidated dataset was de-identified and data on the nature or location of the injury was not accessible for this study.

Fort Jackson currently trains 50% of all individuals entering the US Army's basic training and 60% of all females entering the US Army each year [19]. Although age was not available in the database; basic training recruits are generally between the ages of 17 to 21 years. Individual race/ethnicity information was also not available, however, a 2010 report on overall basic training recruit demographics published total recruit percentages of 63.6% white, 18.9% black, 4.8% Asian, 0.8% Native American, and 11.9% Hispanic [20].

The protocol for this study was determined as not constituting human subjects research by the United States Military Academy Institutional Review Board (#18–020).

## Data preparation

BTRs were provided protocols for scanning, however, recruits were not supervised for correct position while inside the scanner. Therefore, some records were missing measurements or contained physiologically implausible measurements. Images of records that had missing measurements or physiologically implausible measurements confirmed that the BTR within the scanner did not follow proper positioning protocol. We removed any records containing five or more missing measurements (2,481 records) to account for this measurement error. To account for implausible measurements, we removed any records with paired measurements on the left and right side of the body (e.g. left and right leg) differing by more than 2 standard deviations (2,214 records).

The final reference database contained 12,985 (25.1% female) records with 97 of these participants sustaining an injury that resulted in medical separation (0.7%). A workflow diagram similar to the cross-industry standard process for data mining protocol [21] describing the data preparation process appears in Fig 1.

**Balancing the injured and non-injured data.** The ratio of individuals that were injury free to individuals that sustained a medical separation injury in our dataset was approximately 133:1. With such an imbalanced ratio of negative to positive outcomes, a constant model that predicts zero injury would be over 99% accurate. In order to derive meaningful models that predict which individuals are at higher risk for medical separation due to injury we need to (1) amplify the signal from the injured class in order for our models to learn the relevant features, and (2) use measures other than accuracy for evaluating out-of-sample performance.

To address the first concern, we oversampled with replacement of the injured cohort in the training dataset until the ratio was 1:1. This enhances the training process to learn features

Fig 1. Work-flow diagram describing the data preparation to model evaluation process.

associated with the injured cohort observations. The employed balancing process is equivalent, under certain conditions, to setting different penalties on misclassification of the injured vs. non-injured cohort, sometimes termed *class weighting*. We chose to apply the oversampled data for training all models because class weighting in a neural network trained with stochastic

gradient descent can affect the effective learning rate [22]. To address the second concern, evaluating out-of-sample performance, we used a ROC curve, discussed later.

**Feature engineering.** We explored several techniques for reducing the dimensionality and collinearity of the dataset and adding meaningful structure. We first reduced the dimension and collinearity of the data by replacing each paired body measurement (e.g. left and right arm length) with its average value. We then performed a *k*-means clustering [18] of the entire dataset, retaining each record's cluster assignment as an additional explanatory variable using what is referred to as a one-hot encoding scheme [22]. Each observation's assigned cluster membership was included as a feature in the predictive models.

We also evaluated the use of Principal Component Analysis [18, 22] to reduce the dimensionality of our data prior to model training, which we expected to further improve the quality of our models due to the large amounts of collinearity in the data, although we ultimately discarded this approach, as described in Results.

## Predictive models

We evaluated three machine learning predictive models: logistic regression, random forests, and neural networks [18]. We used each model's out-of-sample prediction performance on injuries resulting in medical discharge as a basis for model comparison. Specifically, we trained and tested each model using a stratified *k*-fold cross-validation scheme, described in more detail below, and evaluated performance using each model's mean area under the curve (AUC) of a receiver operating characteristic (ROC) curve. We selected ROC curves to evaluate performance instead of precision-recall curves because they do not rely on the assumption that the out of sample baseline probabilities will be the same as in sample probabilities.

**Model selection and comparison.** There are two sets of parameters determined during the training process: tuning parameters (sometimes termed *hyperparameters* [22]) which are set before training, and all other parameters (sometimes termed *weights* or *coefficients* [22]) learned during training. For example, the logistic regression model consists of a single hyperparameter controlling regularization penalty of model complexity (in our case the L2-norm of the model parameters) and the regression coefficients, one for each feature, including a bias term. We refer to the selection of hyperparameters as *model selection*, and the evaluation of different models' out-of-sample predictive performance as *model comparison*.

Because of the scarcity of injured outcomes, we did not reserve an independent test data set for model evaluation. We instead used average cross-validation scores to measure out-of-sample prediction accuracy for both model selection and comparison.

For model selection, we used randomized search over a grid of possible hyperparameters, with stratified 3-fold cross-validation at each iteration. For each model we began by creating a grid of possible hyperparameters, then iteratively selecting from this grid at random. At each iteration, we evaluated the average out-of-sample performance of the model using the current set of hyperparameters using *k*-fold cross-validation. Specifically, the entire dataset was partitioned randomly into three sets, or *folds* [22], each containing an approximately equal number of injured individuals. We oversampled (with replacement) the injured records in each fold to create a 1:1 ratio of positive and negative outcomes ("stratification"). We used a small number of folds r to retain a reasonable number of injured records per fold and minimize the erratic performance that would result from oversampling 5–10 records hundreds of times. Two of the folds were then used to train the model, with the remaining fold used to test the model, and this process was repeated three times. The ROC AUC was retained for each test fold, along with the confusion matrix corresponding to the threshold closest to the optimal outcome of zero false positives and a true positive rate of one. We repeated this 3-fold stratified cross-

validation procedure twice for each set of hyperparameters, and computed the average ROC AUC over all six runs. After completing 100 iterations of this procedure, we selected the set of hyperparameters with the best average validation score.

For model comparison, we simply compared each model's average AUC under the cross-validation scheme outlined above, using the optimal set of hyperparameters. We did not take into account any qualitative or quantitative aspects of the models apart from this predictive ability out-of-sample. As a baseline model, we used BMI and gender in a logistic regression model, using the same model selection and comparison methodology as above.

We used different but standard approaches to investigate feature importance in each model. For logistic regression, we examined the standardized regression coefficients of each feature. For the random forests, we compared the "variable importance" resulting from a comparison of the number of decision trees in which the variable appears, normalized by the associated node impurity decrease. For the neural network, we compared the normalized weights of the input layer.

## Results

### Participants

Age and race were not available in the database, however, the majority of BTRs are between the ages of 17–22 years old. Participant characteristics in the injured and non-injured classes appear in Table 1. Reported are the breakdowns in male and female cohorts in the original dataset and reference data used for analysis with observations removed for cases with more than five missing measurements or 3 implausible measurements (i.e. left/right measurements differing by more than 2 standard deviations). In both male and female cohort, those discharged from the service were slightly heavier than those who were not.

### Model results

**Feature engineering.** The dimension reduction scheme for averaging paired measurements reduced the total features from 161 to 126 consisting of 125 body measurements and gender. We retained 10 clusters by directly inspecting how much of the cumulative variance is explained by addition of each cluster.

**Table 1. Description of participant characteristics.** The characteristics are provided from the original dataset obtained from the scanner, the reduced dataset after eliminating observations that included more than five missing measurements and the final reference dataset after removing observations with three or more implausible measurements.

| Data | N (%Female) | Injured | BMI (kg/m$^2$) |
|---|---|---|---|
| Original | 17,680 (27.5%) | Males: 147 | Males: 25.08 ± 3.82 |
| | | Females: 74 | Females: 23.73 ± 2.93 |
| | | | Injured: 25.59 ± 3.99 |
| < 5 missing measurements | 15,199 (25.4%) | Males: 125 | Males: 25.12 ± 3.83 |
| | | Females: 63 | Females: 23.65 ± 2.96 |
| | | | Injured: 25.62 ± 3.97 |
| < 5 missing measurements &< 3 implausible measurements | 12,985 (25.1%) | Males: 97 | Males: 25.46 ± 3.70 |
| | | | Females: 23.68 ± 2.80 |
| | | Females: 51 | Injured: 24.31 ± 3.58 |

Data is reported as mean ± SD.

**Table 2. Model AUC, 95% confidence interval and influential variables.**

| Model | AUC 95% CI | Highest weighted model variables[†] |
|---|---|---|
| Logistic Regression | 0.67 [0.62, 0.72] | Head circumference, torso length |
| Random Forest | 0.65 [0.61, 0.70] | Leg length, Torso length, ankle circumference |
| Neural Network | 0.70 [0.68, 0.72] | Torso length |

[†]Influential variables identified using standardized coefficients (logistic regression), node impurity decrease (Random Forest), or mean first layer absolute weight (neural network).

Use of principal components for feature extraction resulted in reduced out-of-sample scores in all models, so we ultimately discarded this in favor of the simpler related approach of averaging paired measurements.

**Predictive models.** The AUC for all final models were above 0.50 (Table 2) with logistic regression AUC = 0.67 [+/- 0.06], random forest AUC = 0.65 [+/- 0.05] and neural network AUC = 0.70 [+/- 0.02] (Fig 2). The neural network outperformed the other models and yielded a smaller variance in AUC (Fig 2D). All models outperformed a baseline model using only BMI and gender, which achieved an AUC = 0.61 [+/- 0.05].

For the neural network, we also report the confusion matrix resulting from summing across all three folds for one iteration of cross-validation (see S1 Table). The overall true positive rate is 69% and false positive rate is 35%, using an average threshold of 0.45.

The hyperparameters and model architectures selected through cross-validation are as follows. For logistic regression, we selected an L2-penalty with strong regularization. For the random forest, we selected 30 base estimators with a maximum of 12 features in use per tree. For the neural network, we selected one hidden layer of 12 neurons, with tanh activation, and strong L2-regularization.

A summary of variables which were most important to each model's predictions, as described in Methods, is given in Table 2. We note torso length appears in all three models, all models appear to rely on non-standard measurements available through the laser scanner, but leave further interpretation for Discussion.

## Discussion

Here, we for the first time utilize body measurements obtained by a 3D body image scanner to predict injuries of BTRs that result in discharge from the US Army during basic combat training. The model correctly classified Soldiers at risk for dischargeable injuries with a true positive rate of 69% and incorrectly classified Soldiers at a false positive rate of 35% (S1 Table). The algorithm was programmed into the 3D body image scanner and can be used at Fort Jackson to identify BTRs at risk in real time during the 20 second scan for uniform fitting offering an opportunity to identify Soldiers at risk for discharge due to injury.

Our work builds upon existing studies that identified risk factors for injury such as gender, low fitness prior to entering basic training, and high or low BMI [23–26]. The findings presented here extend these results by leveraging the predictive accuracy of machine learning techniques and by using improved anthropometric measurements that can be accessed during a 20-second body scan. These advancements lead to personalized predictions that do not require additional measures than what are already being routinely conducted at the site.These findings also strongly suggest that machine learning models using results from a 3D body image scanning could be used to predict injuries in other sports [27, 28] or clinical health outcomes [29].

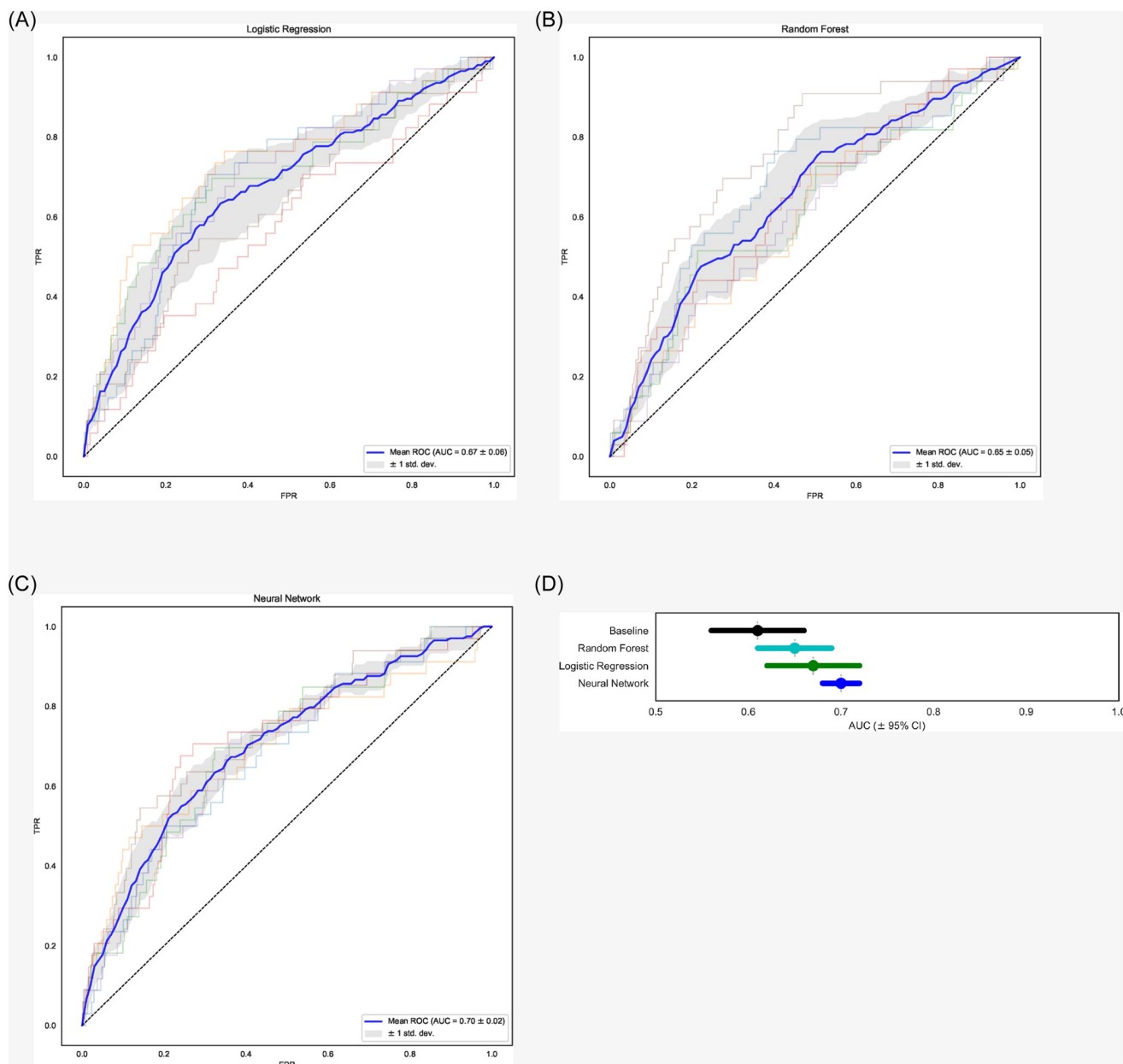

**Fig 2.** ROC curves for logistic regression (Panel A), random forest (Panel B) and neural network (Panel C) models over repeated, stratified, 3-fold cross-validation. Solid curve represents the mean ROC curve. AUC for each model were 0.67 ± 0.05, 0.65 ± 0.04, and 0.70 ± 0.02, respectively. Panel D is a plot of the AUC for each model ± 95% confidence interval.

## Model interpretation

The complexity of machine learning models often enables greater predictive ability, but it also increases the difficulty of interpreting the model itself; for example, quantifying the importance of different input features to the model as in Table 2. Logistic regression is a classic technique with well-accepted measures for quantifying the idea of variable importance in a rigorous way. Random forests and neural networks have more inscrutable inner structure, making interpretation more difficult. Although we present estimates of each model's

influential variables using standard techniques in the field, these calculations by no means imply that only these variables are important, that they are important in the same way, or that we can draw any biomechanical conclusions from them.

## Study limitations

While this study predicted all injuries that resulted in discharge from basic training, data on injury site or type of injury (e.g. situational vs. stress-related) was not available for analysis. With this type of additional information, the physiological mechanisms underlying the injury could be explored further. We expect the correlation between body measurements and injury is greater in stress-related cases, and therefore limiting the data to this subset would increase the predictive ability of the models to detect stress-related injuries.

A potential explanation for our findings could be that a BTR arrives at basic training with a subclinical pre-existing injury that only presents itself during physical activity, which we were unable to investigate due to limited information; however, including 3D body scanned images into the analysis performed in the epidemiology study conducted at the request of Fort Jackson by the U.S. Army Center for Health Promotion and Preventive Medicine [6] could potentially reveal this mechanistic insight.

Another study limitation involves concerns with the measurement records of BTRs that were not positioned correctly in the scanner. We have access to a second database of Human Solutions measurements collected at the Lackland Air Force Base. The measurements from Lackland Air Force Base did not include concomitant outcomes on injury and therefore could not be applied for modeling injury from body shape data. However, the scans at the Lackland Air Force Base were obtained under supervision, and in the 64,000 scans performed at Lackland, there was not a single record containing missing or implausible measurements. This discrepency suggests that to preserve scan quality, the scanner protocols need to be carefully explained and that individuals should be supervised during their scan.

Finally, we relied on the pre-programmed anthropometric measurements integrated in Human Solutions. For future work, the raw data imaged by Human Solutions could be applied to develop additional body site measurements specific to stress-related injuries as model inputs.

Our findings can also be triangulated with earlier results that identified risk factors [3]. These earlier results identified BTRs with low or high BMI who smoked, and were female, were at higher risk for injury. Using a combined classification approach that leverages the earlier results with the body scan derived model stands to improve predictive accuracy beyond what we have found here. The model presented here and a classification algorithm could be programmed directly into the scanner to identify BTRs at risk automatically while they are being scaanned for uniform sizing. Those flagged at risk can be referred to the base athletic trainers who could then tailor training protocols to build strength and stamina in such BTRs.

## Conclusions

Artificial intelligence models that predict outcomes integrated with new technologies like the 3D body scanner provide new scalable and efficient opportunities to minimize injuries. Commanding officers now have a tool that will help them increase the readiness of their units.

## Supporting information

**S1 Table. Confusion matrix for the neural network model, with true positive rate (TPR) is 69.3%, and the false positive rate (FPR) is 35.2%.** This is based on average across all cross-validation test folds using threshold yielding optimal TPR and FPR, and note this is slightly

different than the TPR and FPR reported in the paper which is based on the optimal TPR/FPR of the average ROC curves.
(DOCX)

## Acknowledgments

The authors have no conflicts of interest to declare. The authors have not received funding for this work. DMT and MS developed this study. SM designed the models, performed the analysis, and wrote the first draft of the study. KT, PK and MS performed additional data analysis. SBH and RLA reviewed the analysis and wrote portions of the manuscript. All authors reviewed and revised multiple manuscript drafts.

We would like to thank LTG (ret) Mark Hertling for bringing our attention to this problem. We would also like to thank Robert Bona from Human Solutions, LTC Jason Pieri, MAJ Brian Kriesel, and the Fort Jackson team for their support and assistance in compiling the data. We also appreciate the feedback given by an anonymous reviewer, which greatly improved the manuscript. The views expressed in this work are those of the authors and do not reflect the official policy or position of the United States Military Academy, Department of the Army, or the Department of Defense. The results of the study are presented clearly, honestly, and without fabrication, falsification, or inappropriate data manipulation. The results of the present study do not constitute endorsement by ACSM.

## Author Contributions

**Conceptualization:** Steven Morse, Michael Scioletti, Steven B. Heymsfield, Diana M. Thomas.

**Data curation:** Diana M. Thomas.

**Formal analysis:** Steven Morse, Kevin Talty, Patrick Kuiper, Michael Scioletti, Diana M. Thomas.

**Investigation:** Steven Morse, Diana M. Thomas.

**Methodology:** Steven Morse, Patrick Kuiper, Michael Scioletti.

**Project administration:** Steven Morse, Diana M. Thomas.

**Supervision:** Michael Scioletti, Steven B. Heymsfield, Richard L. Atkinson, Diana M. Thomas.

**Writing – original draft:** Steven Morse, Diana M. Thomas.

**Writing – review & editing:** Steven Morse, Kevin Talty, Patrick Kuiper, Michael Scioletti, Steven B. Heymsfield, Richard L. Atkinson, Diana M. Thomas.

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
