## [Decision Letter · Decision Letter 0]

1 May 2020

PONE-D-20-08676

Machine learning prediction of combat basic training injury from 3D body shape images

PLOS ONE

Dear Dr. Thomas,

Thank you for submitting your manuscript to PLOS ONE. After careful consideration, we feel that it has merit but does not fully meet PLOS ONE’s publication criteria as it currently stands. Therefore, we invite you to submit a revised version of the manuscript that addresses the points raised during the review process.

We would appreciate receiving your revised manuscript by Jun 15 2020 11:59PM. To enhance the reproducibility of your results, we recommend that if applicable you deposit your laboratory protocols in protocols.io, where a protocol can be assigned its own identifier (DOI) such that it can be cited independently in the future. For instructions see: http://journals.plos.org/plosone/s/submission-guidelines#loc-laboratory-protocols

We look forward to receiving your revised manuscript.

Kind regards,

Ulas Bagci, Ph.D.

Academic Editor

PLOS ONE

Additonal Editor Comments:

The paper has some merits, and reviewers have consensus on this.

There are, however several concerns as well regarding the study design and specific research questions.

I recommend authors to prepare a response letter to those questions with a revised manuscript for further consideration.

Journal Requirements:

3. Please ensure that you refer to Figure 3 in your text as, if accepted, production will need this reference to link the reader to the figure.

4. Please include a caption for figure 3.

5. Please include captions for your Supporting Information files at the end of your manuscript, and update any in-text citations to match accordingly. Please see our Supporting Information guidelines for more information: http://journals.plos.org/plosone/s/supporting-information

Reviewers' comments:

Reviewer's Responses to Questions

**Comments to the Author**

1. Is the manuscript technically sound, and do the data support the conclusions?

Reviewer #1: Yes

Reviewer #2: Yes

2. Has the statistical analysis been performed appropriately and rigorously? 

Reviewer #1: Yes

Reviewer #2: Yes

3. Have the authors made all data underlying the findings in their manuscript fully available?

Reviewer #1: No

Reviewer #2: No

4. Is the manuscript presented in an intelligible fashion and written in standard English?

Reviewer #1: Yes

Reviewer #2: Yes

5. Review Comments to the Author

Reviewer #1: * Very cool motivation for the study. The data is preexisting from 3D body scans for uniform fitting so there is not additional cost (from a real-world application point of view) to employ such a method. Overall I like this study but there are a few areas of missing details and baselines which should be addressed before acceptance.

* The authors make a good point about the applicability of body measurements possibly having a high correlation with "overuse injuries", and BMI is known to be a poor metric. However, there is likely little to no correlation with other sports and military injuries such as the authors mentioned femoral neck injuries. In hockey and likely other areas these injuries are highly situational and not about being "out of shape". It would be wise for the authors to separate any sort of repetitive stress injuries from these more situational injuries, the correlation will likely be much higher, but it seems such data was not available to the authors.

* "...total recruit percentages of 18.9% black, 4.8% Asian, 0.8% Native American, and 11.9% Hispanic (22)" This doesn't add to 100%. Please update.

* The authors state that 97 subjects had injuries resulting in separation, but the confusion matrix only shows 33 injuries?

* The description of the ANN model used is not included. Is it a MLP? How many layers?

* A correlation with BMI and injuries would be a nice baseline to justify these 3D scanners are superior to a basic measurement like that. The reviewer thinks such a comparison is pretty crucial to be added. It seems this work was done in previous studies and currently no comparison with previous work is provided. This would provide a nice connection with previous studies.

* With such limited data, why did the authors do 3-fold cross validation? Something like 10-fold is more common. It gives more training data. The authors don't need to redo all the experiments, but it might lead to superior performance (more training data + more powerful ANN can be used to get better performance).

* Possible suggestions on future work: 1) Don't rely on the identified body measurements, directly use the 3D scans and methods from the computer vision community to possibly identify better features than those which are optimal for uniform measurements. 2) The false positive rate is a big concern for applicability. The authors should think of ways to address this to have any chance of real-world application.

Reviewer #2: The authors propose to predict the risk probability of injury due to basic combat training. The authors use the 3D body shape images captured by a device to extract 161 features to describe the subject. Then reduce feature dimension to 126 by averaging and then clustering them. To model these features, the authors use logistic regression, random forest, and a neural network. The NN performs better than the other modeling methods with AUC 0.70. This is an application paper.

Questions and Comments:

Q1

Line 27-30

What is the basis for postulating that certain body characteristics (as captured in the 3D body shape images) can be used to predict the risk of injury due to physical activity?

Q2

Line 40-42

How can you be certain that this is due to the correlation? Do you have features extracted from the 3D body shape images captured before the physical activity to compare?

Q3

Line 48-50

What purpose does it serve? This is a technical paper.

Q4

Line 176

Explain the choice of the hyperparameters used in logistic regression and random forest. The authors note that NN performs better. However, there is no architecture or NN design choices. Please provide.

Q5

Line 213 - 218

Did you apply both averaging and k-means clustering to reduce the feature dimension from 161 to 125 or just used averaging? Please explain clearly how did you get the final features.

Q6

Please provide an ablation study on the NN choice. I suggest the authors use transfer learning to improve performance.

6. PLOS authors have the option to publish the peer review history of their article (what does this mean?). If published, this will include your full peer review and any attached files.

Reviewer #1: No

Reviewer #2: No

---

## [Author Response · Author response to Decision Letter 0]

13 May 2020

PONE-D-20-08676

Machine learning prediction of combat basic training injury from 3D body shape images

Response to Reviewers

Editor Comments:

Response: We have renamed our files as indicated in the URLs and revised the title page as directed.

2. We note that you have indicated that data from this study are available upon request. PLOS only allows data to be available upon request if there are legal or ethical restrictions on sharing data publicly. For information on unacceptable data access restrictions, please see http://journals.plos.org/plosone/s/data-availability#loc-unacceptable-data-access-restrictions. If there are ethical or legal restrictions on sharing a de-identified data set, please explain them in detail (e.g., data contain potentially identifying or sensitive patient information) and who has imposed them (e.g., an ethics committee). Please also provide contact information for a data access committee, ethics committee, or other institutional body to which data requests may be sent.

Response: The data sharing needs to be approved by the Fort Jackson leadership and this requires a memo to be sent to the Chief of Staff of the base. These are Army requirements. We will help anyone who is interested in accessing the data. The Chief of Staff changes every 2-3 years and the contact details will change but in the Data Availability statement, interested parties can be directed to contact the current Chief of Staff at Fort Jackson to receive command approval.

3. Please ensure that you refer to Figure 3 in your text as, if accepted, production will need this reference to link the reader to the figure.

Response: There are only two Figures. This was an oversight on our part.

4. Please include a caption for figure 3.

5. Please include captions for your Supporting Information files at the end of your manuscript, and update any in-text citations to match accordingly. Please see our Supporting Information guidelines for more information: http://journals.plos.org/plosone/s/supporting-information

Reviewer #1:

Comment 1: Very cool motivation for the study. The data is preexisting from 3D body scans for uniform fitting so there is not additional cost (from a real-world application point of view) to employ such a method. Overall I like this study but there are a few areas of missing details and baselines which should be addressed before acceptance.

Response: We thank the reviewer for their comment. We also found this project interesting and enjoyable. 

Comment 2: The authors make a good point about the applicability of body measurements possibly having a high correlation with "overuse injuries", and BMI is known to be a poor metric. However, there is likely little to no correlation with other sports and military injuries such as the authors mentioned femoral neck injuries. In hockey and likely other areas these injuries are highly situational and not about being "out of shape". It would be wise for the authors to separate any sort of repetitive stress injuries from these more situational injuries, the correlation will likely be much higher, but it seems such data was not available to the authors.

Response: The reviewer is correct, our intent was to demonstrate the predictive ability of body measurements for injury in a population of relatively sedentary individuals, and we agree it is more reasonable to expect correlation with stress-related injuries, not situational injury. Moreover, because the recruits are not likely elite athletes, injury in hockey players or other elite athletes are very different. Unfortunately, our data did not include on which injuries were overuse injuries and which were situational. We clarified this in the introduction by removing suggestions that our work will be transferrable to other sports.

Revised Paragraphs in the Introduction:

The United States Army (US Army) anticipates basic training recruits (BTRs) will be injured during training. Most of these injuries will heal with rest, but some are more severe, leading to expensive treatments and medical separation from the US Army (1). Injuries, such as stress fractures (2, 3), can result in discharges from the US Army basic combat training. Femoral neck injuries are also observed during basic training and can often result in discharge (4). 

 These injuries, some of which are situational and others of which are stress-related (or “overuse injuries”(5)), have high medical costs and can impact future quality of life for the BTR. Additionally, under the US Army’s medical policy, the US Army may be financially responsible for the injured BTR’s long-term care. Thus identifying BTRs at increased risk for dischargeable injury prior to training is critical (6-8). 

Revised Paragraph in the Discussion:

Study Limitations

While this study predicted all injuries that resulted in discharge from basic training, data on injury site or type of injury (e.g. situational vs. stress-related) was not available for analysis. With this type of additional information, the physiological mechanisms underlying the injury could be explored further. In particular, we expect the correlation between body measurements and injury is greater in stress-related cases, and therefore limiting the data to this subset would increase the predictive ability of the models to detect stress-related injuries. 

Comment 3: "...total recruit percentages of 18.9% black, 4.8% Asian, 0.8% Native American, and 11.9% Hispanic (22)" This doesn't add to 100%. Please update.

Response: The remaining 63.6% of recruits are white, we have updated the paper to reflect this and the revision appears below for the reviewer’s convenience:

Individual race/ethnicity information was also not available, however, a 2010 report on overall basic training recruit demographics published total recruit percentages of 63.6% white, 18.9% black, 4.8% Asian, 0.8% Native American, and 11.9% Hispanic (22).

Comment 4: The authors state that 97 subjects had injuries resulting in separation, but the confusion matrix only shows 33 injuries?

Response: The confusion matrix only depicted the result in one test fold – we agree this is not the full picture, and have amended to show the total results over all test folds. 

The amended confusion matrix and caption appears below:

Table S1. Confusion matrix for the neural network model, with true positive rate (TPR) of 69.1% and false positive rate (FPR) of 34.9%. This is based on the totals across all cross-validation test folds, using threshold yielding optimal TPR and FPR.

 Predicted

 Non-injured Injured

Actual Non-injured 8,384 4,504

 Injured 30 67

Comment 4: The description of the ANN model used is not included. Is it a MLP? How many layers?

Response: The ANN selected through cross-validation was a MLP with a single hidden layer of 12 neurons and tanh activation. We used the scikit-learn implementation of the ADAM algorithm for training. We have included this description in the manuscript which has been pasted below.

For the neural network, we selected one hidden layer of 12 neurons, with tanh activation, and strong L2-regularization.

Comment 5: A correlation with BMI and injuries would be a nice baseline to justify these 3D scanners are superior to a basic measurement like that. The reviewer thinks such a comparison is pretty crucial to be added. It seems this work was done in previous studies and currently no comparison with previous work is provided. This would provide a nice connection with previous studies.

Response: We have included a more detailed explanation of motivation in the introduction. We were indeed motivated exactly by BMI studies. Our interest in this topic originated after hearing a presentation on BMI and injuries at Fort Jackson by LTG (ret) Mark Hertling. We we have also included the results of a baseline model using logistic regression with only gender and BMI in the manuscript and in Figure 2D.

The revision to the introduction appears below:

Low physical fitness, low and high body mass index (BMI), and anthropometry are some of the well-established predictors of Army related injuries (8, 11-16). While, for example, BMI defined categories demonstrated risk, the number of false positives identified by high or low BMI would be too high to be a feasible screening mechanism. Screening BTRs with more manually obtained measurements is not currently feasible, in part due to the added burdens of measuring predictive model input variables and then delivering model predictions quickly and efficiently for the high volume of BTRs that continuously arrive at basic training sites like Fort Jackson, SC.

The revision to the methods on the baseline BMI model comparison:

As a baseline model, we used BMI and gender in a logistic regression model, using the same model selection and comparison methodology as above.

The revision to the results on the BMI baseline model in comparison to the 3D body image data models:

Predictive models

The AUC for all final models were above 0.50 (Table 2) with logistic regression AUC = 0.67 [+/- 0.06], random forest AUC = 0.65 [+/- 0.05] and neural network AUC = 0.70 [+/- 0.02] (Figure 2). The neural network outperformed the other models and yielded a smaller variance in AUC (Figure 2 D). All models outperformed a baseline model using only BMI and gender, which achieved an AUC = 0.61 [+/- 0.05].

Figure 2D:

Comment 6: With such limited data, why did the authors do 3-fold cross validation? Something like 10-fold is more common. It gives more training data. The authors don't need to redo all the experiments, but it might lead to superior performance (more training data + more powerful ANN can be used to get better performance).

Response: We agree 5- or 10-fold cross-validation would be preferable, but we chose 3-fold due to the extreme sparseness of positive cases. We added wording to clarify this choice in the methods:

Revision to Methods:

For model selection, we used randomized search over a grid of possible hyperparameters, with stratified 3-fold cross-validation at each iteration. For each model we began by creating a grid of possible hyperparameters, then iteratively selecting from this grid at random. At each iteration, we evaluated the average out-of-sample performance of the model using the current set of hyperparameters using k-fold cross-validation. Specifically, the entire dataset was partitioned randomly into three sets, or folds (24), each containing an approximately equal number of injured individuals. We oversampled (with replacement) the injured records in each fold to create a 1:1 ratio of positive and negative outcomes (“stratification”). We used a small number of folds in order to retain a reasonable number of injured records per fold and minimize the erratic performance that would result from oversampling 5-10 records hundreds of times.

Comment 7: Possible suggestions on future work: 1) Don't rely on the identified body measurements, directly use the 3D scans and methods from the computer vision community to possibly identify better features than those which are optimal for uniform measurements. 2) The false positive rate is a big concern for applicability. The authors should think of ways to address this to have any chance of real-world application.

Response: We agree with the reviewer. Human Solutions is less agreeable for us to use the raw data, however, we have begun to work with Styku. This company shares all internal data with us and we already have done better in our prediction models using this data. Also the Human Solutions device is not portable. 

Off the record, we programmed the model into Human Solutions and applied it to the next round of BTRs and the model identified only 11 BTRs at risk. Those BTRs were given stabilizing exercises and that round of training ended with no discharge injuries. We cannot publish this because the Army has not and will not likely approve this as a study, but the Battalion Commander was reluctant to sit on the information the model yielded. He felt he needed to protect his Soldier’s. 

Since we cannot discuss this exercise in a publication, we provided it in the discussion as future work:

The model presented here and a classification algorithm could be programmed directly into the scanner to identify BTRs at risk automatically while they are being scaanned for uniform sizing. Those flagged at risk can be referred to the base athletic trainers who could then design specific preventative training protocols.

The suggestion by the reviewer to work with the raw data was included in the study limitations:

Finally, we relied on the pre-programmed anthropometric measurements integrated in Human Solutions. For future work, the raw data imaged by Human Solutions could be applied to develop additional body site measurements specific to stress-related injuries as model inputs.

Reviewer #2:

The authors propose to predict the risk probability of injury due to basic combat training. The authors use the 3D body shape images captured by a device to extract 161 features to describe the subject. Then reduce feature dimension to 126 by averaging and then clustering them. To model these features, the authors use logistic regression, random forest, and a neural network. The NN performs better than the other modeling methods with AUC 0.70. This is an application paper.

Questions and Comments:

Q1

Line 27-30

What is the basis for postulating that certain body characteristics (as captured in the 3D body shape images) can be used to predict the risk of injury due to physical activity?

Response: This is a great question and was also asked by the other reviewer. Our interest in this topic originated after hearing a presentation on BMI and injuries at Fort Jackson by LTG (ret) Mark Hertling. BMI is an anthropometric measurement and our team asked LTG (ret) Hertling if we could take a portable device to Fort Jackson to scan Soldiers. We were surprised to find out that Fort Jackson already had a scanner which is where this research took off.

We have now included a more detailed explanation of motivation in the introduction. We were indeed motivated exactly by BMI studies. As per the request of Reviewer 1, we closed the loop to include the results of a baseline model using logistic regression with only gender and BMI for comparison in the manuscript and in Figure 2D.

The revision to the introduction appears below:

Low physical fitness, low and high body mass index (BMI), and anthropometry are some of the well-established predictors of Army related injuries (8, 11-16). While, for example, BMI defined categories demonstrated risk, the number of false positives identified by high or low BMI would be too high to be a feasible screening mechanism. Screening BTRs with more manually obtained measurements is not currently feasible, in part due to the added burdens of measuring predictive model input variables and then delivering model predictions quickly and efficiently for the high volume of BTRs that continuously arrive at basic training sites like Fort Jackson, SC.

The revision to the methods on the baseline BMI model comparison:

As a baseline model, we used BMI and gender in a logistic regression model, using the same model selection and comparison methodology as above.

The revision to the results on the BMI baseline model in comparison to the 3D body image data models:

Predictive models

The AUC for all final models were above 0.50 (Table 2) with logistic regression AUC = 0.67 [+/- 0.06], random forest AUC = 0.65 [+/- 0.05] and neural network AUC = 0.70 [+/- 0.02] (Figure 2). The neural network outperformed the other models and yielded a smaller variance in AUC (Figure 2 D). All models outperformed a baseline model using only BMI and gender, which achieved an AUC = 0.61 [+/- 0.05].

Figure 2D:

Q2

Line 40-42

How can you be certain that this is due to the correlation? Do you have features extracted from the 3D body shape images captured before the physical activity to compare?

Response: Correlated is not that right word because we are not using statistical inference. A better word would be predicts. We have revised these lines in the manuscript and the revision appears below:

Body shape profiles generated from a three-dimensional body scanning imaging in military personnel predicted dischargeable physical injury. The ANN model can be programmed into the scanner to deliver instantaneous predictions of risk, which may provide an opportunity to intervene to prevent injury.

Q3

Line 48-50

What purpose does it serve? This is a technical paper.

Response: We agree and we have removed it.

Q4

Line 176

Explain the choice of the hyperparameters used in logistic regression and random forest. The authors note that NN performs better. However, there is no architecture or NN design choices. Please provide.

Response: Hyperparameters for all three models were chosen through stratified cross-validation. We have now included much more detailed explanations for the choices for the models. The revision appears below:

Model Selection and Comparison

 There are two sets of parameters determined during the training process: tuning parameters (sometimes termed hyperparameters(24)) which are set before training, and all other parameters (sometimes termed weights or coefficients (24)) learned during training. For example, the logistic regression model consists of a single hyperparameter controlling regularization penalty of model complexity (in our case the L2-norm of the model parameters) and the regression coefficients, one for each feature, including a bias term. We refer to the selection of hyperparameters as model selection, and the evaluation of different models’ out-of-sample predictive performance as model comparison. 

Because of the scarcity of injured outcomes, we did not reserve an independent test data set for model evaluation. We instead used average cross-validation scores to measure out-of-sample prediction accuracy for both model selection and comparison.

For model selection, we used randomized search over a grid of possible hyperparameters, with stratified 3-fold cross-validation at each iteration. For each model we began by creating a grid of possible hyperparameters, then iteratively selecting from this grid at random. At each iteration, we evaluated the average out-of-sample performance of the model using the current set of hyperparameters using k-fold cross-validation. Specifically, the entire dataset was partitioned randomly into three sets, or folds (24), each containing an approximately equal number of injured individuals. We oversampled (with replacement) the injured records in each fold to create a 1:1 ratio of positive and negative outcomes (“stratification”). We used a small number of folds in order to retain a reasonable number of injured records per fold and minimize the erratic performance that would result from oversampling 5-10 records hundreds of times. Two of the folds were then used to train the model, with the remaining fold used to test the model, and this process was repeated three times. The ROC AUC was retained for each test fold, along with the confusion matrix corresponding to the threshold closest to the optimal outcome of zero false positives and a true positive rate of one. We repeated this 3-fold stratified cross-validation procedure twice for each set of hyperparameters, and computed the average ROC AUC over all six runs. After completing 100 iterations of this procedure, we selected the set of hyperparameters with the best average validation score.

For model comparison, we simply compared each model’s average AUC under the cross-validation scheme outlined above, using the optimal set of hyperparameters. We did not take into account any qualitative or quantitative aspects of the models apart from this predictive ability out-of-sample. 

Q5

Line 213 - 218

Did you apply both averaging and k-means clustering to reduce the feature dimension from 161 to 125 or just used averaging? Please explain clearly how did you get the final features.

Response: We first used averaging of paired measurements to reduce the feature dimension to 126 (including gender), then the result of k-means clustering (with k = 10 clusters) to increase the feature dimension to 136. We have clarified these portions of data preparation in the manuscript in two places in the methods. These paragraphs are pasted below for the reviewer’s convenience.

Data Preparation

BTRs were provided protocols for scanning, however, recruits were not supervised for correct position within the scanner. Therefore, some records were missing measurements or contained physiologically implausible measurements. Images of records that had missing measurements or physiologically implausible measurements confirmed that the position of BTR within the scanner did not follow proper protocol. To eliminate any record that did not follow scanner protocols, we removed any records containing five or more missing measurements (2,481 records). We also removed any records with paired measurements on the left and right side of the body (e.g. left and right leg) differing by more than 2 standard deviations (2,214 records). 

The final reference database contained 12,985 (25.1% female) records with 97 participants sustaining an injury that resulted in medical separation (0.7%). A workflow diagram similar to the cross-industry standard process for data mining protocol (23) describing the data preparation process appears in Figure 1. 

Feature Engineering

We explored several techniques for reducing the dimensionality and collinearity of the dataset and adding meaningful structure. We first reduced the dimension and collinearity of the data by replacing each paired body measurement (e.g. left and right arm length) with its average value. We then performed a k-means clustering(20) of the entire dataset, retaining each record’s cluster assignment as an additional explanatory variable using what is referred to as a one-hot encoding scheme (24). Each observation’s assigned cluster membership was included as a feature in the predictive models. 

We also evaluated the use of Principal Component Analysis (20, 24) to reduce the dimensionality of our data prior to model training, which we expected to further improve the quality of our models due to the large amounts of collinearity in the data, although we ultimately discarded this approach, as described in Results.

Q6

Please provide an ablation study on the NN choice. I suggest the authors use transfer learning to improve performance.

Response: We thank the reviewer for this suggestion and agree that ablation studies are extremely important in deep learning architectures, but since our NN is quite small by comparison (single hidden layer of 12 neurons) and heavily regularized during training, we do not feel it would be a fruitful topic of investigation. Similarly, while we agree transfer learning is a powerful method to increase performance in certain contexts, we do not have access to an appropriate pre-training dataset in this study. It is possible pre-training on a simulated dataset, or on a different target, may aid in weight initialization or feature extraction, but we believe it is more appropriate to leave this to future wor

---

## [Decision Letter · Decision Letter 1]

8 Jun 2020

Machine learning prediction of combat basic training injury from 3D body shape images

PONE-D-20-08676R1

Dear Dr. Thomas,

We’re pleased to inform you that your manuscript has been judged scientifically suitable for publication and will be formally accepted for publication once it meets all outstanding technical requirements.

Kind regards,

Ulas Bagci, Ph.D.

Academic Editor

PLOS ONE

Additional Editor Comments (optional):

A successful rebuttal period,

and reviewers found the article to be an important application paper.

Please note that reviewers also mentioned that authors need to clearly mention in the article that the data will be available upon request with appropriate contact email/address.

Please proofread as well.

Reviewers' comments:

Reviewer's Responses to Questions

**Comments to the Author**

1. If the authors have adequately addressed your comments raised in a previous round of review and you feel that this manuscript is now acceptable for publication, you may indicate that here to bypass the “Comments to the Author” section, enter your conflict of interest statement in the “Confidential to Editor” section, and submit your "Accept" recommendation.

Reviewer #1: All comments have been addressed

Reviewer #2: All comments have been addressed

2. Is the manuscript technically sound, and do the data support the conclusions?

Reviewer #1: (No Response)

Reviewer #2: Yes

3. Has the statistical analysis been performed appropriately and rigorously? 

Reviewer #1: (No Response)

Reviewer #2: N/A

4. Have the authors made all data underlying the findings in their manuscript fully available?

Reviewer #1: (No Response)

Reviewer #2: No

5. Is the manuscript presented in an intelligible fashion and written in standard English?

Reviewer #1: (No Response)

Reviewer #2: Yes

6. Review Comments to the Author

Reviewer #1: Thank you for addressing all of my concerns. Congratulations on your revisions and your great work.

Reviewer #2: Thank you for addressing my questions and comments. One final note on the data availability. The authors note that they will help "anyone" who is interested in accessing the data. And also, state that the data needs to be requested and "authorized" from the United States Army. I request the authors to address who can request, share, and for what purpose.

7. PLOS authors have the option to publish the peer review history of their article (what does this mean?). If published, this will include your full peer review and any attached files.

Reviewer #1: No

Reviewer #2: No

---

## [Editor Report · Acceptance letter]

17 Jun 2020

PONE-D-20-08676R1 

Machine learning prediction of combat basic training injury from 3D body shape images 

Dear Dr. Thomas:

I'm pleased to inform you that your manuscript has been deemed suitable for publication in PLOS ONE. Congratulations! Your manuscript is now with our production department. 

Kind regards, 

on behalf of

Dr. Ulas Bagci 

Academic Editor

PLOS ONE